# Thyroid Lobectomy for Low to Intermediate Risk Differentiated Thyroid Cancer

**DOI:** 10.3390/cancers12113282

**Published:** 2020-11-06

**Authors:** Dana M. Hartl, Joanne Guerlain, Ingrid Breuskin, Julien Hadoux, Eric Baudin, Abir Al Ghuzlan, Marie Terroir-Cassou-Mounat, Livia Lamartina, Sophie Leboulleux

**Affiliations:** 1Department of Surgery, Anesthesia and Interventional Medicine Gustave Roussy, 94805 Villejuif, France; joanne.guerlain@gustaveroussy.fr (J.G.); ingrid.breuskin@gustaveroussy.fr (I.B.); 2Department of Nuclear Medicine and Endocrine Oncology, Anesthesia and Interventional Medicine Gustave Roussy, 94805 Villejuif, France; julien.hadoux@gustaveroussy.fr (J.H.); eric.baudin@gustaveroussy.fr (E.B.); marie.terroir-cassou-mounat@gustaveroussy.fr (M.T.-C.-M.); livia.lamartina@gustaveroussy.fr (L.L.); sophie.leboulleux@gustaveroussy.fr (S.L.); 3Department of Biology and Pathology, Anesthesia and Interventional Medicine Gustave Roussy, 94805 Villejuif, France; abir.alghuzlan@gustaveroussy.fr

**Keywords:** thyroid cancer, lobectomy, thyroidectomy, prognosis, risk stratification

## Abstract

**Simple Summary:**

Total thyroidectomy used to be recommended for all thyroid cancers. We now know that some thyroid cancers have a relatively low risk of recurrence. Today, for some of these cancers, depending on the type of tumor, its’ size and other tumor characteristics, a thyroid lobectomy (or hemithyroidectomy) can be performed without increasing the patient’s risk of cancer recurrence. Thyroid lobectomy has the advantages of having less risk of surgical complications and a less frequent need for thyroid hormone replacement therapy. This approach is not optimal for all thyroid cancers, however, and careful tumor and patient selection are necessary. This review explains the rationale and criteria for patient selection for thyroid lobectomy for selected thyroid cancers.

**Abstract:**

Many recent publications and guidelines have promoted a “more is less” approach in terms of treatment for low to intermediate risk differentiated thyroid cancer (DTC), which comprise the vast majority of thyroid cancers: less extensive surgery, less radioactive iodine, less or no thyroid hormone suppression, and less frequent or stringent follow-up. Following this approach, thyroid lobectomy has been proposed as a means of decreasing short- and long-term postoperative morbidity while maintaining an excellent prognosis for tumors meeting specific macroscopic and microscopic criteria. This article will examine the pros and cons of thyroid lobectomy for low to intermediate risk cancers and discuss, in detail, criteria for patient selection and oncological outcomes.

## 1. Introduction

Many recent publications and guidelines have promoted a “more is less” approach in terms of treatment for low-risk differentiated thyroid cancer (DTC), which comprises the vast majority of thyroid cancers: less extensive surgery, less radioactive iodine, less or no thyroid hormone suppression, and less frequent or stringent follow-up [1,2]. This has been due to several factors. First, we now realize that in most developed countries, there has been too much thyroid surgery performed over the past decades, and there is a backlash from physicians and patients alike. A study for over one year, including almost the entire database of the French healthcare system, found that out of 35,367 thyroid surgeries performed, only 82% of patients had had a thyroid ultrasound (US) in the year preceding surgery and only 34% had had fine-needle aspiration cytology [3]. A recent study in the USA found that the currently recommended thyroid nodule care pathway in terms of thyroid imaging was not followed in 38% of cases [4,5]. Following this patient pathway would decrease the number of non-necessary thyroid surgeries for the benefit of patients and healthcare systems alike.

Secondly, many studies have now confirmed the long-term complications and consequences of thyroid surgery for patients and healthcare systems despite the use of modern technologies such as neuromonitoring. The rates of unilateral and bilateral recurrent nerve paralysis and the rate of permanent hypoparathyroidism remain stable and low at 1–2%, but this is responsible for significant morbidity from a patient perspective and is very costly on the scale of entire populations [6,7,8,9].

Finally, the concept of personalized medicine, extended to thyroid cancer, implies tailoring the degree of surgery to the oncological risk. Data on overall survival for low-risk tumors show low to nonexistent disease-specific mortality for these tumors. The evaluation of the risk–benefit ratio between less and more surgery is based on the risks of total thyroidectomy in terms of morbidity and quality of life versus the risk of disease recurrence.

This article will examine the pros and cons of thyroid lobectomy for low to intermediate risk cancers, with the exclusion of very low risk intrathyroidal micropapillary thyroid carcinoma.

## 2. What Are Low-, Intermediate-, and High-Risk Cancers?

In 2009, the guidelines of the American Thyroid Association (ATA) established a classification using pre- and postoperative tumor characteristics to predict the outcomes of therapy in terms of the risk of disease recurrence. These guidelines classified differentiated thyroid cancer as having a low, intermediate, or high risk of recurrence according to tumor size, extrathyroidal extension, completeness of resection, final pathology, vascular invasion, metastatic lymph nodes, and distant metastases [10]. The most recent 2015 ATA Guidelines proposed to include the evaluation of even more disease characteristics to further refine the risk of disease recurrence [1]. The new criteria for risk stratification include lymph node metastases size, extranodal extension, the extent of vascular emboli, and the presence of BRAF and/or TERT mutations (Table 1). Tumors are considered at high risk of recurrence when any of the following characteristics are present: gross extrathyroidal extension, grossly incomplete resection, lymph node metastases of >3 cm, the presence of TERT mutation, follicular thyroid cancer with >4 vascular emboli, the presence of distant metastases, or a thyroglobulin suggestive of distant metastases; these tumors are associated with an estimated recurrence risk of 40% or more. The rate of recurrence for low-risk disease, that is, an intrathyroidal tumor with complete resection, no vascular emboli, no aggressive histologic features, less than 5 micrometastases <2 mm in lymph nodes, and no distant metastases, is estimated to be less than or equal to 5%. For intermediate-risk disease, with microscopic extrathyroidal extension and/or vascular invasion (for papillary cancer), and/or aggressive histology, and/or more than 5 lymph node metastases (all <3 cm or clinical N1), the rate of locoregional recurrence was estimated to be between 5% and 30%.

The TNM classification is used to determine disease-specific mortality [11,12,13]. Cancers classified as T1 T2 N0 M0, that is, intrathyroidal tumors or those with microscopic extrathyroidal extension, have an estimated disease-specific survival at 20 years of 99% [14]. Patients <55 years old, without distant metastases, even with metastatic lymph nodes (N1a and N1b, TNM Stage I) have an estimated 10-year disease-specific survival of 98–100%. For patients >55 years old, the 10-year disease-specific survival rate is 98–100% for Stage I disease, T1 T2 N0 M0, and 85–95% for Stage II disease (T1 T2 N1 M0 or T3 N0 N1 M0) [11,13].

Given these very high disease-specific survival rates, therapy for these low- and intermediate-risk cancers aims, then, not only to achieve optimum disease-specific survival but also disease-free survival, that is, to minimize the risk of tumor recurrence. Early TNM stage disease can be associated with a low, intermediate, or high risk of recurrence, and, as a consequence, both the TNM and ATA risk classifications should be considered for appropriate management of differentiated thyroid carcinoma. The ATA risk classification gives more information on the risk of recurrence and the TNM classification more on death from the disease [15]. Microscopic extrathyroidal extension is thus considered to be a risk factor for locoregional recurrence but is no longer considered a risk factor for disease-specific mortality; the TNM T3 classification now only includes macroscopic extrathyroidal extension. Aggressive histology, even for small tumors, is considered to be a factor for an intermediate risk of recurrence and includes tall cell, hobnail, columnar cell, and diffuse sclerosing papillary carcinomas [16]. For follicular carcinoma, low-risk tumors include the encapsulated variant, minimally invasive follicular carcinoma, and encapsulated minimally invasive follicular carcinoma with a small number of foci of vascular invasion and without extracapsular vascular invasion [1]. Follicular carcinoma with a large number of vascular emboli and/or extracapsular vascular invasion, even in encapsulated follicular carcinomas, are considered to be in the high-risk category due to an increased risk of distant metastases [1].

Finally, ongoing risk stratification further allows for the prediction of recurrence in an individual patient that has undergone surgery with radioactive iodine. Risk of recurrence is based on serum Tg and imaging (ultrasound and, if performed, whole-body scan after RAI administration), with an “excellent response”—defined as serum Tg < 0.2 ng/mL (or TSH-stimulated Tg < 1 ng/mL) and negative imaging—being associated with a 1–2% risk of recurrence; there is the exception of high-risk patients, who have a residual risk of 14% even if they achieve “excellent response” status at some point during their follow-up [1,17,18] (Table 2). The response to therapy criteria was subsequently revised to incorporate follow-up and prognosis after thyroid lobectomy [19]. In their multicenter retrospective study of 187 patients treated with thyroid lobectomy, 22 patients received completion surgery (11.8%). Forty percent of patients in this study had thyroid microcarcinoma T1 a. Lobectomy-treated patients with an “excellent response”—a stable or declining nonstimulated serum Tg level <30 ng/mL, negative serum anti-Tg antibodies, and negative imaging (ultrasound)—during the first two years of follow-up had no structural evidence of disease during an average of 8–11 years of subsequent follow-up. Patients with an “indeterminate response”—nonspecific imaging and persistent, stable, or declining anti-Tg antibodies—had a 4.3% rate of structural recurrence. Patients with a “biochemical incomplete response”—negative imaging but rising Tg or Tg-antibody levels—had a 33.3% rate of structural recurrence (Table 2). The value of Tg after lobectomy has been further evaluated in more recent retrospective studies and will be discussed under the heading “Long Term Follow-up” below.

## 3. Why Thyroid Lobectomy?

The aim of choosing thyroid lobectomy or lobectomy plus isthmusectomy (hemithyroidectomy) for thyroid cancers is to optimize oncologic outcomes, such as disease-specific and disease-free survival, while, at the same time, keeping surgical morbidity and long-term effects of treatment to a strict minimum.

Why prefer a lobectomy to a total thyroidectomy? The complication rate of total thyroidectomy is higher than that of thyroid lobectomy, even for experienced surgeons. In a large meta-analysis of 50,445 patients, Kandil et al. found a pooled relative risk for postoperative complications to be 10.67 for total thyroidectomy versus thyroid lobectomy [20]. The authors found a relative risk of 1.69 for temporary recurrent nerve paralysis, 1.85 for permanent recurrent nerve paralysis, and 2.58 for hematoma. The relative risk for temporary hypocalcemia was 3.1 and for permanent hypocalcemia 1.69.

In a large database study of 62,722 hospitalizations in the USA, the overall complication rate for lobectomy for high-volume surgeons (>99 cases per year) was 7.6%, whereas the complication rate for total thyroidectomy was 14.5% [21]. These numbers increased to 9.9% and 18.8%, respectively, for intermediate-volume surgeons (10–98 cases per year) and 11.8% and 24.1%, respectively, for low-volume surgeons performing less than 10 thyroidectomies per year.

For thyroid surgery in general, the complication rate decreases with the surgeon’s experience, as reflected by the number of thyroid surgeries performed per year. Studying a national database in the USA, Adam et al. found that of 16,954 patients undergoing total thyroidectomy, 81% of patients were operated on by surgeons performing less than 26 thyroidectomies annually. The median number of thyroidectomies per year per surgeon was 7, and 51% of surgeons performed less than 1 thyroidectomy per year. Taking the complication rate of surgeons performing over 26 thyroidectomies per year as the gold standard, the authors found that the complication rate increased almost proportionally as the number of thyroidectomies per year decreased, with 22% more complications for surgeons performing 11–15 surgeries per year, 68% more complications for those performing 2–5 per year, and 87% more complications for those performing 1 or less per year [22]. Similarly, Kandil et al. have shown a global complication rate of 18.9% for surgeons practicing less than 10 thyroidectomies per year, decreasing to 13.4% for those performing 10–100 cases per year, and to 7.5% for surgeons performing over 100 thyroidectomies per year [23]. The fact that over 80% of patients in the USA are treated by surgeons performing less than 26 cases per year is one factor that has pushed guidelines toward less extensive initial surgery in an attempt to avoid undue complications in low- and intermediate-risk patients.

Thyroid hormone supplementation is required after total thyroidectomy, but after thyroid lobectomy, many patients may not need supplementation. The overall rate of hypothyroidism after lobectomy, as analyzed in a recent meta-analysis pooling 51 studies, was 30% [24]. Risk factors for supplementation after lobectomy include lymphocytic thyroiditis, antithyroid antibodies, low remnant lobe volume, higher preoperative TSH levels, and lower preoperative free T4 levels [24,25]. For Kandil et al., analyzing 15,412 patients in a different meta-analysis, a preoperative TSH level >2.5 mUI/L was associated with a relative risk of 3.52 for postoperative hypothyroidism. The presence of thyroid antibodies conferred a relative risk of 3.5 for hypothyroidism, and the presence of thyroiditis on the pathology report conferred a relative risk of 3.17 [20]. In a cohort study of almost 9000 patients undergoing partial thyroidectomy in France in 2010, 44% were taking thyroid hormones after 12 months following surgery [3]. Taking into account the 2015 ATA guidelines that recommend target TSH levels in the range of 0.5–2 mUI/L after lobectomy for patients with thyroid cancer, however, up to 73% of patients—excluding those with thyroiditis—may need thyroid hormone supplementation [26].

The advantages of thyroid lobectomy, combined with the plethora of data confirming the excellent survival of patients with low-risk cancers treated with lobectomy, has led to a recent shift towards lobectomy. Furthermore, lobectomy has been shown to be a more cost-effective approach to treating a hypothetical 2.5-cm papillary thyroid carcinoma, even with completion thyroidectomy in the case of initially unrecognized high-risk features (aggressive histology, microscopic extrathyroidal extension, lymphovascular invasion, positive resection margin, nodal metastases, and multifocality) [27].

With regard to the impact of the extent of surgery on long-term health-related quality of life (HRQoL), only limited and conflicting evidence is available to date. In a survey study using two validated questionnaires to investigate global HRQoL, functional and symptom scores (European Organisation for Research and Treatment of Cancer Quality of Life Core Questionnaire version 3.0, Thyroid Cancer-Specific Questionnaire Module version 2.0) comparing 211 patients who underwent total thyroidectomy versus 59 who underwent lobectomy alone, Bongers et al. found no significant difference in quality-of-life scores [28]. An additional questionnaire on cancer-related anxiety (Assessment of Survivor Concerns) was administered to the same cohort and revealed a significantly higher concern about recurrence in the lobectomy patients compared with the total thyroidectomy patients. The patients were surveyed from 1 to more than 10 years after their thyroid surgery. Another survey study of 1005 patients, 695 of whom had had total thyroidectomy, was undertaken by means of a structured telephone interview [29]. Approximately two-thirds of the patients complained of at least one HRQoL issue, and the extent of surgery was associated with a significatively higher risk of HRQoL change vs. no change. HRQoL was 1.5-times worse in patients having undergone total thyroidectomy (95% CI, 1.04–2.12) and 2.3-times worse in those having undergone total thyroidectomy and neck dissection (95% CI, 1.10–4.80) compared with patients having undergone lobectomy [29]. Finally, a recent online survey asking 150 patients to state their choice between total thyroidectomy and thyroid lobectomy found that their choices were, on average, determined for 35% of patients by the risk of cancer recurrence, 28% by the need for a second operation, 19% by the risk of recurrent nerve damage, and 9% by the risk of hypocalcemia and need for thyroid hormone treatment [30]. Oncologic considerations thus seem to be more important to patients than complications, and we will focus on this aspect of thyroid lobectomy in the following paragraphs.

## 4. Oncologic Considerations: Completion Thyroidectomy and Survival

The need for completion thyroidectomy is determined by the oncological risks: risk of cancer recurrence, the indication for adjuvant radioiodine therapy or ablation, and the need to have a reliable and sensitive tumor marker—serum thyroglobulin (Tg)—to detect recurrences. Historically, total thyroidectomy as the initial procedure was considered to be safer than lobectomy, followed by completion thyroidectomy in terms of postoperative complications and in terms of completeness of resection of the thyroid tissue [31]. Today, however, it has been shown in many retrospective studies that completion thyroidectomy does not carry more morbidity than initial upfront total thyroidectomy and that the completeness of resection, as measured by postoperative serum Tg, is comparable as well [32,33,34,35]. In fact, several studies have even demonstrated that the rate of temporary postoperative hypoparathyroidism may be lower for completion thyroidectomy [34,35]. The question of the safety of completion lobectomy is, therefore, today, no longer an issue. This reassuring data is also an argument in favor of the strategy of conservative surgery or “staged surgery” for 1–4 cm tumors, reserving completion thyroidectomy for patients with higher-risk factors on final histopathology [36].

From an oncological standpoint, analyzing overall survival, a study including 61,775 patients from the National Cancer Institute Database suggested that lobectomy was sufficient for differentiated thyroid cancer <4 cm, without evidence of lymph node metastases, without microscopic extrathyroidal extension, and without multifocality: overall survival in this subgroup of patients was comparable between lobectomy and total thyroidectomy [37]. Of note, however, in that study, 88% of patients had undergone total thyroidectomy and only 12% had undergone a lobectomy. 

In the most recent 2015 ATA Guidelines, this large study constituted a high level of evidence in favor of thyroid lobectomy due to the absence of a difference in survival between lobectomy and total thyroidectomy for low-risk cancers [38,39,40,41,42]. The change in criteria for thyroid lobectomy between the 2009 and 2015 ATA guidelines led to a significant change in the proportion of patients undergoing total thyroidectomy versus lobectomy but also in the number of patients requiring completion thyroidectomy. In the study by Hiroshen et al., initial total thyroidectomy was performed in 61% of patients prior to the 2015 ATA guidelines but only in 31% of patients following the publication of these guidelines, a statistically significant decrease [43]. The rate of completion thyroidectomy decreased significantly, in that study, from 73.9% to 20%. However, other studies have found higher rates for the need for completion lobectomy. According to several retrospective studies, multifocality can be observed in 15–63% of cancers of 1–4 cm [44,45] and microscopic extrathyroidal extension in up to 66% of these tumors [46,47]. If we follow the strict criteria of lobectomy only for unifocal tumors without microscopic extrathyroidal extension and without lymph node metastases or other intermediate or high-risk features (see above), 34–59% of patients undergoing initial lobectomy will require completion thyroidectomy due to finding intermediate- or high-risk features upon final pathology of the lobectomy specimen [48,49,50,51,52,53,54,55].

Even if preoperative patient selection is very careful and careful intraoperative evaluation excludes patients with suspected extrathyroidal extension or lymph node metastases, at least 30% of patients may still require completion thyroidectomy secondarily due to higher-risk features that can only be ascertained upon definitive histopathological and molecular analysis of the tumor [56,57]. Patient preference, recently studied via an online questionnaire, found that among other considerations, patients favored lobectomy over total thyroidectomy as long as the risk of needing a completion operation was below 30% [30]. Unfortunately, patient preferences do not fall within the estimated 34–59% rate of completion thyroidectomy, as cited above. In the systematic review by Vargas-Pinto et al., if minimal microscopic positive margins and microscopic extrathyroidal extension were excluded from the criteria for requiring completion thyroidectomy, the estimated rate of the need for completion thyroidectomy, across the included studies, dropped to 11% [48].

## 5. Oncologic Considerations: Recurrence in the Contralateral Lobe and Lymph Node Metastases

Taking into account that the tumors treated only with lobectomy do indeed meet the histopathological criteria for lobectomy—1 to 4 cm, unifocal, without microscopic extrathyroidal extension or microscopic lymph node metastases—cancer recurrence rates in the range of 2.7–11.8% have been reported. In a recent meta-analysis of 175,430 patients, Bojoga et al. reported a 7% rate of cancer recurrence [58]. This large meta-analysis found no difference in recurrence rates for intrathyroidal cancers up to 4 cm in diameter without lymph node metastases, treated with lobectomy or with total thyroidectomy. A further systematic review of 31 studies and 228,746 patients found that recurrence rates after thyroid lobectomy were slightly but significantly higher than after total thyroidectomy: 9% versus 7.4%, *p* = 0.001) [59]. Cancer recurrence in the contralateral lobe was observed in 4.8% of patients in the retrospective study by Ritter et al. of 167 patients followed for an average of 6.5 years [60], with an additional 2.4% of patients with recurring lymph node metastases. Matsuzu et al. reported a 6.5% rate of recurrence in the remaining thyroid and a 9.4% rate of recurrence in regional lymph nodes in 1088 patients followed for 25 years [38]. On the other hand, Lee et al. reported a 0.9% recurrence rate in the contralateral lobe, but 87% of the tumors in this study were micropapillary cancers [61]. Many retrospective studies, with various inclusion criteria, have reported global recurrence rates ranging from 4.2–7.1% after lobectomy for low-risk cancers [19,62,63,64,65]. Contralateral thyroid nodules generally appear within five years of initial therapy [66]. For cancer recurrence after lobectomy, a study by Matsuzu et al. found that the majority of recurrences appeared within the first 10 years, with few recurrences appearing later during their 25-year follow-up period [38]. For low-risk, cN0 cancers treated with total thyroidectomy, prophylactic neck dissection, and RAI, one study found that most structural recurrences requiring therapy appeared within the first 6.5 years of follow-up, with few late recurrences [67]. A larger study of more than 1000 mainly low to intermediate risk patients found that all of the recurrences occurred within 8 years of initial therapy [68]. A relatively long follow-up is thus necessary for these patients, and retrospective studies lacking follow-up may underestimate the true incidence of recurrence.

Historically, prophylactic lymph node dissection of the central compartment was a mainstay in the surgical management of thyroid cancers, along with total thyroidectomy. Some advantages to systematically performing prophylactic neck dissection, such as decreased central compartment recurrence rates, lower incidences of retreatment, and lower postoperative thyroglobulin levels, have been found in retrospective studies, but with a low level of evidence and with only an objectively small difference in the range of 2–7% [67,69,70,71,72,73,74,75]. On the other hand, many other retrospective studies have shown disadvantages—a higher risk of temporary hypoparathyroidism—or at least no advantages to performing neck dissection in the absence of metastatic lymph nodes detected upon preoperative ultrasound [76,77,78,79,80,81,82,83,84,85]. Today, in light of these studies and with recent improvements in high-resolution ultrasound and fine-needle aspiration cytology techniques for lymph nodes, systematic prophylactic central neck dissection is suggested only for high-risk cancers, although some centers may still use prophylactic neck dissection as a staging tool for low-to-intermediate risk cancers [1]. Several prospective randomized studies are currently underway to finally put this debate to rest (www.clinicaltrials.gov), but the results are still pending. Macroscopic exploration of the central compartment during the thyroid surgery has not been shown to be superior to high-resolution ultrasound for detecting metastatic lymph nodes and, thus, is not currently recommended [86].

Thyroid lobectomy with ipsilateral prophylactic central neck dissection is routinely performed by some high-volume surgical groups for low-risk cancers. A recent meta-analysis of this practice, including 14 studies and 688 patients, showed a significant decrease in the rate of cancer recurrence as compared to lobectomy alone [87]. The recurrence rate was 1.3% in the lobectomy plus neck dissection group versus 5.4% in the lobectomy alone group. As seen in the case of central neck dissection accompanying total thyroidectomy, this difference, although statistically significant, was clinically very small: only 4.1% of patients actually benefited from this practice, with no benefit for 95.9% of patients.

## 6. Oncologic Considerations: Need for Radioactive Iodine

Radioactive iodine (RAI) administration is generally only considered to be effective once a total thyroidectomy has been performed. One objective of RAI is to ablate the remnant of normal thyroid tissue remaining in the neck in order to obtain a very reliable and sensitive measure of serum thyroglobulin, which guides prognostication and follow-up (see below). The other use of RAI is therapeutic: with higher doses and, in some cases, with thyroid hormone withdrawal to stimulate endogenous TSH, a therapeutic effect on suspected residual disease—in the neck in the form of microscopic lymph node metastases (adjuvant RAI) or in the form of macroscopic known residual disease or distant metastases (therapeutic RAI)—is expected [88].

RAI is currently to be “considered” and “generally favored” in the 2015 ATA Guidelines for intermediate-risk cancers with microscopic extrathyroidal extension and central or lateral lymph node metastases, with the goal to reduce the risk of persistent or recurrent disease in the neck [1]. When these intermediate-risk factors are present on postoperative pathological analysis after a lobectomy, it is thus “generally favored” to perform completion thyroidectomy in order to administer RAI. The decision to administer RAI, however, may be modulated according to the degree of microscopic extrathyroidal invasion, the number and size of the metastatic nodes, the availability of high-quality neck ultrasonography, the postoperative status of the neck, and, particularly, the specialized risk evaluation, as ascertained by the treating team and the willingness and ability of the patient to submit to regular follow-up in a specialized center [1,88]. The risk of not performing RAI in terms of locoregional recurrence must be weighed against the expertise of the team to detect structural recurrences but also against the patient’s anxiety level in terms of risk of recurrence, possible surgical complications, thyroid hormone supplementation and need for specialized follow-up [88]. In a retrospective cohort study of 1000 patients with 1–4 cm differentiated thyroid cancer, 19.5% of those initially eligible for thyroid lobectomy would have required completion thyroidectomy based on indications for adjuvant RAI alone [49].

For low-risk cancers, the imaging studies obtained after thyroid remnant ablation with RAI do not add any further information about residual disease or risk of recurrence as compared to follow-up with ultrasound and thyroglobulin (Tg) [89], and RAI does not seem to add any advantage in terms of survival or risk of recurrence for these tumors with an already low recurrence rate [90,91]. The most recent ATA guidelines, therefore, do not recommend routine administration of RAI for low-risk patients [1]. 

## 7. Long Term Follow-Up and Detection of Recurrences: Thyroglobulin (Tg)

Serial dosing of serum thyroglobulin (Tg) after therapy is, today, the most sensitive means for follow-up and detection of disease recurrence [1]. Using Tg for follow-up after thyroid lobectomy remains debatable, however, because the remaining thyroid lobe may produce significant amounts of Tg. In addition, given that the rate of recurrence is very low for these low-risk cancers after lobectomy, close and precise follow-up using Tg is maybe not necessary for these patients [92].

Momesso et al. were among the first to define a cutoff for Tg levels associated with an excellent response to therapy after lobectomy [19]. They defined an excellent response after lobectomy as the combination of Tg <30 ng/mL, negative anti-Tg antibodies, and negative neck imaging. A biochemically incomplete response was defined as serum Tg >30 or rising and/or increasing antibodies, with negative imaging. Recurrent/persistent structural disease was found in 6.4% of the 187 patients in their cohort treated with lobectomy, 18% of which were initially considered as “intermediate risk.” Structural recurrence was observed in 33% of patients with a biochemically incomplete response versus 0% of patients with an excellent response after lobectomy. The authors concluded that the 30 ng/mL cutoff was pertinent for follow-up after lobectomy.

More recent studies, however, have questioned the 30 ng/mL cutoff. In the cohort study by Park et al., 208 patients with low-risk cancers, treated with lobectomy and not requiring thyroid hormone replacement therapy, were followed for a median of 6.9 years [93]. For all patients, Tg levels tended to increase over time, with an average of a 10% increase annually. Nine percent of patients recurred, but there was no significant difference in Tg levels or in the rate of increase in Tg levels between patients in remission and patients with recurrence. In fact, patients without recurrence were more likely to have Tg increase >20% over time. Another cohort study of 167 patients also observed a gradual increase in Tg levels over time [60]. In the absence of anti-Tg antibodies, average postoperative Tg levels were 14.7 ± 19.0 ng/mL. They observed 7.2% recurrences over a 6.5-year follow-up after lobectomy, but Tg levels, Tg dynamics, and anti-Tg antibody dynamics were not predictive of or correlated with recurrence. Cho et al. retrospectively evaluated 619 patients treated with lobectomy, of which 45% were considered by ATA as intermediate risk. The recurrence rate was 6.5%: 1.5% in the ATA low-risk group and 5% in the intermediate-risk group [94]. Using the response to therapy/dynamic risk stratification criteria, as defined by Momesso et al. [19], an “excellent response” after lobectomy was associated with a 1.6% structural recurrence rate, a “biochemically incomplete response” was associated with a 2.9% recurrence rate and an “indeterminate response” with a 3.8% recurrence rate. The difference in recurrence rates among these 3 different response categories was not statistically significant, but the dynamic risk stratification criteria were an adjuvant to the ATA classification in predicting recurrences. Vaisman et al. also observed that for 72 patients treated with lobectomy, Tg levels were not predictive of recurrence, which occurred in 4.2% of cases [64].

Recurrences for low-risk cancers after lobectomy occur primarily in the contralateral lobe and neck lymph nodes, so in light of this data showing that Tg levels and fluctuations in Tg alone cannot be interpreted as being in favor of cancer recurrence, follow-up should primarily be based on ultrasound [92,95].

## 8. Preoperative Patient Counseling

As seen above, many factors are involved when deciding between performing an initial lobectomy or an initial total thyroidectomy for thyroid cancer (Table 3). Preoperative patient counseling needs to address all of these issues: postoperative complications, postoperative thyroid hormone substitution, the risk of requiring completion thyroidectomy and possibly RAI, the risk of cancer recurrence, and follow-up modalities and frequency. Added to this complexity are variations in patients’ levels of anxiety and fear of cancer recurrence and death. Patient worry was recently explored by surveying 2215 disease-free thyroid cancer survivors, 2 to 4 years after treatment [96]. Among those patients, almost two-thirds (63.2%) reported persistent worry about cancer recurrence. A majority also reported worry about family being at risk of cancer (58%) and impairment of quality of life (54.7%). In addition, a large proportion of patients worried about harm from treatment (43.5%) and death (41%). Patients under 44 years of age, females, those without a college degree, Hispanics, and Asians reported significantly more worry. The same group studied physicians’ reactions to patients’ expressions of worry [97]. Out of 448 doctors responding to the survey—endocrinologists, general surgeons, and otolaryngologists—97% answered that when confronted with a worried patient, they proposed to be available to the patient for discussion. Forty-four percent also referred patients to educational websites, 18% encouraged discussion with family and friends, 13% suggested support groups, and 7% referred patients to counselors. When confronted with a “very worried” patient, physicians were more inclined to encourage patients to seek aid outside of the physician–patient relationship. Patients’ worries about cancer recurrence, of course, are not necessarily related to the true risk according to known risk factors. In a recently published survey of 1597 patients, 24.7% overestimated the risk of recurrence and 12.5% overestimated the risk of death [98]. Again, there were demographic disparities, with Hispanic patients and patients without a college education reporting higher distortions of their perception of risk. Overestimation of risk was also related to a higher level of worry and a lower physical quality of life. Patients’ psychological profiles also affect the number of hospital visits and use of medical resources, with patients considered as medical “maximizers” being more inclined to overuse medical visits and imaging studies [99].

Given the complexity of patients’ perceptions of disease and their worries, sensitive and practical tools are needed to correctly measure patients’ worries and needs. A Canadian group has been developing a questionnaire to aid in better understanding individual patients’ concerns preoperatively [100,101]. The “Western Surgical Concern Inventory—Thyroid” questionnaire addresses surgery-related concerns, psychosocial concerns, and daily-living concerns. Areas of major concern in patients undergoing thyroid surgery for a suspicious nodule were noted as follows: having a cancerous nodule, change in voice, surgical complications, need for a second operation, and pain [100]. The second evaluation of this questionnaire also found that the main areas of concern were surgical—voice change, pain, surgical complication, and need for lifelong medication—but patients were also concerned about resuming work and social activities, the risk of becoming depressed, and the risk of becoming a burden on others [101]. These types of questionnaires may aid preoperative physician–patient dialog when addressing specific concerns.

A recent survey of 1124 thyroid cancer survivors conducted by the American Head and Neck Society Endocrine Section found that for 46% of patients, the experience of therapy was not “on par” with their expectations. Seventy percent of them had received printed preoperative instructions, but when asked how they would have liked to have had those instructions delivered, 62.5% responded that they would have preferred a printed pamphlet and 13.7% responded that they would have preferred an online website for the information [102]. Patients particularly requested more attention to the management of energy levels (61%), psychological well-being (50%), and weight changes (48%).

Further studies are warranted to develop practical and pertinent tools for evaluating individual patients’ concerns and needs preoperatively. We also need to focus on evaluating which actions are most helpful to alleviate worry and to adjust patients’ perceptions of risk of recurrence and death to correlate with the excellent prognosis of these low- and intermediate-risk tumors.

## 9. Future Perspectives

Technology and knowledge are rapidly evolving, and new data will certainly change our practices in the near future. Active surveillance, without surgery, is now a justified management option for thyroid microcarcinoma. Recent data on active surveillance for low-risk 1–2 cm tumors are encouraging, and in the future, if further studies confirm the safety of this approach, selected patients with stable T1 bN0 disease may also be eligible for this type of specialized management [103]. Minimally invasive local therapies using radiofrequency ablation, high-frequency ultrasound, and cryotherapy are being employed for the destruction of early-stage nonthyroid malignancies in liver and lung and for benign thyroid lesions. Future studies on their use for the complete eradication of low-risk thyroid cancers may show a role for these technologies in selected cases.

Molecular analysis of thyroid cytology is becoming more and more available and financially accessible [104]. Preoperative molecular analysis today can aid in decision-making for the diagnosis of follicular and indeterminate lesions (Bethesda cytology classes IV and III). With progress in the future, the determination of molecular markers for excellent prognosis may aid in selecting low-risk cancers for active surveillance, with surgery for tumors with markers for poorer prognosis [105,106]. Higher-risk lesions, such as those harboring BRAF and TERT mutations, may benefit from more extensive surgery [107,108]. Molecular markers for RAI uptake or response to redifferentiation protocols could also guide initial surgery for molecularly higher-risk lesions, with total thyroidectomy enabling therapy with RAI if needed [109].

Finally, progress in detecting circulating tumor cells, circulating tumor DNA, or exosomal micro RNA could, one day, be employed for surveillance in thyroid cancer and may be even more sensitive and reliable than Tg, particularly in the setting of Tg antibodies or after lobectomy, obviating the need for total thyroidectomy and/or RAI [110,111,112,113].

## 10. Conclusions

Today, initial lobectomy for low to intermediate risk thyroid cancers of 1–4 cm represents an improvement in patient care, with a lower risk of complications while ensuring a low risk of recurrent disease and cancer-specific mortality. The preoperative work-up is essential to eliminate factors of intermediate risk of recurrence, such as extrathyroidal extension, multifocality, and lymph node metastases. Despite careful work-up, completion thyroidectomy may be required in 30–50% of cases to improve recurrence-free survival. In the future, it is hoped that preoperative molecular analysis may improve patient selection for lobectomy or total thyroidectomy, according to prognostic markers, and innovations in detecting tumor DNA or RNA in the bloodstream may obviate the need for Tg assays and ultrasound.

## Figures and Tables

**Table 1 cancers-12-03282-t001:** Risk groups according to the 2015 American Thyroid Association (ATA) Guidelines [1]. Risk of structural disease recurrence in patients without structural disease after initial therapy.

Risk Group	Tumor Characteristics	Estimated Recurrence Rate %
	Follicular cancer with >4 foci of vascular invasion	30−55%
	T4a tumor with invasion of local structures	30−40%
	Extranodal extension in >3 lymph nodes	40%
	TERT mutation, tumor >1 cm	>40%
	Metastatic lymph node >3 cm	30%
	BRAF mutation + extrathyroidal extension	10−40%
**High Risk**	Papillary carcinoma with vascular invasion	15−30%
	Papillary thyroid cancer with vascular invasion	16−30%
	Clinical N1 or >5 metastatic lymph nodes	20%
	BRAF mutation without extrathyroidal extension and tumor <4 cm	10%
	Microscopic extrathyroidal extension	3−9%
**Intermediate Risk**	Aggressive histology	Varies with tumor size and other histopathological and molecular features
	Microscopic or minor extrathyroidal extension	3−8%
	Up to 5 metastatic nodes	5%
	Any number of metastatic nodes but all <0.2 mm	5%
	24 cm intrathyroidal papillary carcinoma	5%
	Multifocal micropapillary carcinoma	4−6%
	T1 without microscopic extrathyroidal extension and up to 3 metastatic lymph nodes	2%
	Minimally invasive follicular carcinoma	2−3%
	T1 T2 intrathyroidal, BRAF wild-type	1−2%
	Intrathyroidal micropapillary carcinoma BRAF mutated	1−2%
**Low Risk**	Unifocal micropapillary carcinoma	1−2%

**Table 2 cancers-12-03282-t002:** Risk of structurally recurrent/persistent disease according to initial ATA risk classification and ongoing risk stratification.

	Risk of Structurally Recurrent/Persistent Disease				
**ATA Risk Group**	**Response after Therapy**	**TT + RAI**		**TT**	**Lobectomy**
		**Tuttle et al. [17] (*n* = 588; Median Follow-Up 7 Years) ***	**Vaisman et al. [18] (*n* = 425; Median Follow-Up 5 Years) ***	**Momesso et al. [19] ** (*n* = 320; Average Follow-Up 8 Years)**	**Momesso et al. [19] *** (*n* = 187; Average Follow-Up 8 Years)**
	**Excellent**	2% (1/59)	0% (0/96)	0% (0/53)	0% (0/120)
	**Indeterminate**		11.1% (2/18)	0% (0/254)	4.3% (2/46)
	**Biochemically incomplete**	0% (0/30)	35.7% (10/28)	0% (0/7)	33.3% (6/18)
**Low Risk**	**Structurally incomplete**	13% (2/15)	77.7% (7/9)	100% (6/6)	100% (4/4)
	**Excellent**	2% (2/86)	2.6% (2/76)	−	−
	**Indeterminate**		26.7% (4/15)	−	−
	**Biochemically incomplete**	0% (0/56)	55.3% (21/38)	−	−
**Intermediate Risk**	**Structurally incomplete**	41% (41/99)	81.6% (40/49)	−	−
	**Excellent**	14% (2/14)	0% (0/5)	−	−
	**Indeterminate**		25% (1/4)	−	−
	**Biochemically incomplete**	0% (0/9)	80% (12/15)	−	−
**High Risk**	**Structurally incomplete**	79% (81/103)	76.3% (55/72)	−	−

* Best response at two years. ** 91.4% low risk + 8.6% intermediate risk. *** 81.9% low risk + 18.1% intermediate risk.

**Table 3 cancers-12-03282-t003:** Conservative surgery for low to intermediate risk cancers—advantages and disadvantages.

Advantages	Disadvantages
Lower surgical risks versus total thyroidectomy	Risk of completion surgery to improve prognosis and/or administer radioactive iodine
Patient may not require thyroid hormone supplementation	No improvement in long term quality of life has been demonstrated yet
Survival is comparable to total thyroidectomy	Thyroglobulin may not be relevant for follow-up
Completion surgery, if necessary, does not increase surgical risk as compared to total thyroidectomy	Follow-up requires reliable ultrasound
	Not adapted for most intermediate- and high-risk patients
	Patients may still require thyroid hormone supplementation, particularly if thyroiditis or small contralateral lobe

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
