# Peer review of "Thyroid Lobectomy for Low to Intermediate Risk Differentiated Thyroid Cancer"

_cancers, 2020, doi:10.3390/cancers12113282_

Round 1

Reviewer 1 Report

Dear editor and authors,
Thank you for the opportunity to review this paper “Manuscript ID: cancers-969593 Conservative Surgery for Low to Intermediate Risk Differentiated Thyroid Cancer”.
The authors review the pros and cons of “thyroid lobectomy” [ defined as “conservative surgery”] for low to intermediate risk differentiated thyroid cancers and discuss in detail criteria for patient selection and oncological outcomes, including several important issues such as oncologic considerations (Completion Thyroidectomy, Survival, Recurrence in the contralateral lobe/ lymph node metastases, and Need for Radioactive Iodine), and Long term follows up and detection of recurrences (Thyroglobulin).
The submission is original and fits the section and the special issue "2020 Update on the Management of Thyroid Cancer" of this Journal.
The topic is innovative and of interest to the readership, in particular for both Head and Neck/ENT/General/Thyroid surgeons and endocrinologists
Minor Comments 1. “page 4 of 15, line 114 to 11: Conservative surgery is defined as performing a thyroid lobectomy without central neck dissection for thyroid cancers….”. 
It would be better if the authors can provide some references ( better in guideline or consensus paper) for this definition. The use of “Conservative surgery” to represent “thyroid lobectomy without neck dissection” seems to be a new term in the thyroid literature. As the author mentioned in “page 7 of 15, line 282 to 283: Thyroid lobectomy with ipsilateral prophylactic central neck dissection (pCND) is routinely performed by some high-volume surgical groups for low-risk cancers”, it’s true that many surgeons including me use “monitored thyroid lobectomy with routine ipsilateral pCND” as a “Conservative surgery” compare to Total thyroidectomy for low-risk DTC. In addition, for solitary isthmic papillary thyroid carcinoma, the “Conservative surgery” may also refer to isthmusectomy +/- pCND, as compare to lobectomy+ isthmusectomy or total thyroidectomy. Therefore, it would be better if the term “Conservative surgery” in the TITLE/Abstract and Text could be changed to “Conservative surgery with initial lobectomy” or“Conservative lobectomy” to avoid confusion to the readership.
2. Although this paper provides an excellent review that “Conservative surgery with initial lobectomy” today represents an improvement in patient care with a lower risk of complications, patients still have to be counseled about the potential higher risk of recurrence in the contralateral lobe or lymph node, and completion thyroidectomy may be required in 30-50% of cases to improve recurrence-free survival. This issue is important and of interest to the readership. It would be great if the authors can include a section to discuss and review the current literature for preoperative counseling, informed consent, as well as shared decision-making process for choosing “Conservative Surgery for Low to Intermediate Risk DTC”.

Author Response

Dear Reviewer, Dear Editor,

Thank you for taking your valuable time to review our manuscript and for your positive remarks.

As suggested, we have changed the title of the manuscript to read “Thyroid Lobectomy” instead of “conservative surgery”. Some authors use the term “conservative surgery” for “thyroid nodulectomy” (Nahm et al Am J Otolaryngol 2019) and you are absolutely right that the term may be too vague and confusing. We have modified the term in the text accordingly (highlighted in yellow).

Also as suggested, we have added a section about preoperative counseling citing studies of patients’ preferences and preoccupations (new section 8, highlighted in yellow in the text).

Thank you for your thoughtful suggestions for improving this manuscript.

Sincerely,

Dana Hartl

Reviewer 2 Report

This manuscript gives a nice and detailed overview of the arguments pro and contra limited surgery for low to intermediate risk well differentiated thyroid cancer. Although no original data are used, the reviewed literature is representative for this subject, analysed properly and discussed extensively.

The tables represent and summarize the accompanying text. Readability of the tables could be increased by left-aligning the columns instead of centering the text. The third table should be numbered accordingly.

Author Response

Dear Colleague, Dear Editor,

Thank you for taking your valuable time to review our manuscript and for your positive remarks.

We have modified the tables to left-align the columns as you have suggested and labelled Table 3 correctly.

Thank you again for your consideration.

Sincerely,

Dana Hartl

Reviewer 3 Report

This review investigates the role of conservative surgery for low to intermediate risk cancers, with the exclusion of very low risk intrathyroidal micropapillary thyroid carcinoma. The Authors conclude that conservative surgery with initial lobectomy for low to intermediate risk thyroid cancers 1-4 cm represents an improvement in patient care with a lower risk of complications while ensuring a low risk of recurrent disease and cancer-specific mortality, even if completion thyroidectomy may be required in 30-50% of cases to improve recurrence-free survival.

The review is interesting and well-written. I suggest minor points:

  1. Some English and typographical errors are present, please check, see for example: “extracapuslar vascular invasion, …” in line 84; “Data on overall survival for low risk tumors shows …” in line 37; “Of note, however, in this study that 88% of patients had undergone a total thyroidectomy and that only 12% had undergone a lobectomy” in lines 208-210; “…so in light of this data showing …” in line 357
  2. In lines 376-378 “Higher-risk lesions, such as those harboring BRAF and TERT mutations may benefit from more extensive surgery. Molecular markers for RAI uptake or response to redifferentiation protocols could also guide initial surgery for molecularly higher-risk lesions, with total thyroidectomy enabling therapy with RAI if needed”, I suggest to cite recently published papers, such as Gland Surg. 2019 Jun;8(3):298-300

Author Response

Dear Reviewer, Dear Editor,

Thank you for taking your valuable time to review our manuscript and for your positive remarks.

We have corrected necessary grammar and typographical errors (highlighted in yellow in the text). We have added 3 references (references 107, 108, 109), including Gland Surgery 2019, as suggested. They are highlighted in yellow in the text.

Thank you again for your consideration.

Sincerely,

Dana Hartl